# Influence of the Interactions at the Graphene–Substrate Boundary on Graphene Sensitivity to UV Irradiation

**DOI:** 10.3390/ma12233949

**Published:** 2019-11-28

**Authors:** Dorota Nowak, Marian Clapa, Piotr Kula, Mariusz Sochacki, Bartlomiej Stonio, Maciej Galazka, Marcin Pelka, Dominika Kuten, Piotr Niedzielski

**Affiliations:** 1Institute of Materials Science and Engineering, Lodz University of Technology, 1/15 Stefanowskiego St., 90-924 Lodz, Poland; marian.clapa@p.lodz.pl (M.C.); piotr.kula@p.lodz.pl (P.K.); piotr.niedzielski@p.lodz.pl (P.N.); 2Advanced Graphene Products Sp. z o.o., 4 Nowy Kisielin–A. Wysockiego St., 66-002 Zielona Góra, Poland; m.galazka@agp-corp.com (M.G.); m.pelka@agp-corp.com (M.P.); d.kuten@agp-corp.com (D.K.); 3Institute of Micro-and Optoelectronics, Warsaw University of Technology, Koszykowa 75, 00-662 Warszawa, Poland; m.sochacki@imio.pw.edu.pl (M.S.); b.stonio@imio.pw.edu.pl (B.S.)

**Keywords:** graphene sensors, UV irradiation, carrier concentration, wettability, Raman spectroscopy

## Abstract

Graphene is a very promising material for electronics applications. In recent years, its sensitivity to ultraviolet (UV) irradiation has been studied extensively. However, there is no clear answer to the question, which factor has a key influence on the sensitivity of graphene to UV. In order to check the influence of the final substrate on the electrical response, graphene transferred on polymeric and non-polymeric substrate was investigated. To achieve this goal three polymeric and three non-polymeric substrates were tested. The results of the preliminary tests indicated the different character of the reaction on UV irradiation in each of group. To explain the reason of the difference, the complementary studies were done. The samples that were resistant to high temperature were annealed in a vacuum at 500 °C to get rid of water trapped between graphene and the substratum. The samples after annealing reacted less dynamically to UV irradiation. Moreover, the progress of changes in electrical response of the annealed samples had a similar character to the polymeric substrates, with the hydrophobic nature of the surface. These studies clearly prove that the sensitivity of graphene to UV irradiation is influenced by water trapped under the graphene.

## 1. Introduction

The dynamic technological development observed in recent decades has contributed to increase the efficiency of electronic devices reducing their size. As a result, conducted research on 2D materials has increased. One of the most popular in this group of materials is graphene—single layer of sp^2^—hybridized carbon atoms arranged in honeycomb lattice. Due to the fact that 2D materials are only the surface, they are extremely sensitive. Graphene can interact with both the substrate on which it is placed and the surrounding atmosphere. Due to the promising electrical properties of graphene, there have been attempts to use it for the construction of various types of sensors, for example: gas, X-ray [1,2,3]. Another research has proved that graphene is sensitive to UV irradiation [4,5]. But the mechanism of the reaction has still not been explained. Even though there have been many attempts to explain which interactions have a dominant influence on the electrical response of graphene when exposed to UV irradiation, but the opinions are still divided. Some researchers think that the graphene UV sensitivity is connected with desorption of water or gas particles (e.g., NH_3_, NO_2_, O_2,_) from graphene [6,7,8], the second group believe that there are interactions with the final substratum [5]. Graphene is most often tested as a conductive channel of the FET (Field Effect Transistor) transistor, which is deposited on a Si/SiO_2_ substrate [9]. There are no literature reports showing the influence of substratum on the electrical response of graphene during UV irradiation.

The authors of this study investigated the reactions of graphene transferred on six different substrates to UV irradiation. The Raman spectroscopy revealed that only the region with monolayer of graphene strongly reacts to UV irradiation. The reaction of multilayer to UV was almost nonvisible. The contact angle measurement pointed that the polymeric substrates are more hydrophobic. The results of the preliminary tests indicated that polymeric materials are less sensitive to UV irradiation, and the character of reaction to UV is different. The complementary studies unambiguously confirm that the water trapped between graphene and substrate has the superior influence on the sensitivity of graphene to UV irradiation.

## 2. Materials and Methods

High Strength Metallurgical Graphene (HSMG) produced on a bimetallic substrate [10] was used for the study. Graphene was transferred to the target substrate using the wet method. The growth substrate was covered with a thin layer of polymethyl methacrylate (PMMA). Then, the bimetallic substrate was removed by etching in 1 M solution of FeCl_3_. Before a graphene transfer, the target substrates were cleaned in acetone, and the gold electrodes were sputtered. The tests were performed on six different substrates belonging to two groups: polymer (kapton, poly(ethylene terephthalate) (PEN) and poly (tetrafluoroethylene) PTFE and non-polymer (Si/SiO_2_ (280 nm), Al_2_O_3_ and quartz glass). The active surface area of graphene was 3 × 3 mm.

Before UV irradiation a set of tests was done to characterize the substrates as well as the graphene quality after transfer. Water contact angle measurements for all used substrates were performed using the FM40 Easy Drop system with Drop Shape Analysis software (Krüss GmbH). Each measurement was performed on cleaned surfaces such as prepared before transfer. The sample was mounted on the system table, then a 30 μL droplet of deionized water was placed on its surface and contact angle measurement was taken instantly. Graphene quality on different substrates was examined using scanning electron microscope (Hitachi S-3000 N, Japan). Raman spectroscopy (inVia Reflex Renishaw spectrometer, United Kingdom) with an excited wavelength of 532 nm was used to characterize the graphene before and after irradiation on Si/SiO_2_ substrate. The power of the laser on the surface of the examined samples was 10 mW and the applied magnification was 100×. All measurements were carried out in ambient conditions and at room temperature.

UV irradiation was carried out at ambient temperature, at atmospheric pressure in air. Samples were irradiated with a UV diode with a wavelength of 270 nm, impulse irradiation at 50% completion time was used. The distance between the UV source and the sample was 3 cm. During the experiment, the resistance changes progress over time was recorded. Comparative measurements of carrier concentration and mobility were determined by measuring the Hall effect in the electrode system according to the Van der Pauw method. The measurements were carried out in a magnetic field with an induction of 400 mT at a sample current of 1 mA.

## 3. Results and Discussion

In the wet method of transfer, water is used as a medium positioning graphene on the target substrate. Because of that, substrates’ wettability has a key impact on the quality of the transfer. The polymeric substrates used for the research were characterized by a contact angle exceeding 76°, and the ceramics had the wetting at a similar level. Low contact angle values were noticed for silicon substrates: 46° and 57° (Figure 1.). Previous research showed that SiO_2_ contains silane groups and adsorbs water on the surface. On the one hand, this allows for better positioning of graphene on the surface; on the other hand, it traps water between graphene and the substrate [11,12]. Polymer substrates and ceramics were characterized by lower wettability that caused difficulties in the transfer process and could have generated the additional mechanical damage of graphene.

To show the state of the substrates surface and the quality of graphene after the transfer, SEM photos were taken. The research reveals that Si/SiO_2_, quartz glass, kapton and PEN are characterized by a smooth surface. The other two substrates have porosities, but in the case of ceramics, the number of pores is much higher. Figure 2 shows pictures of two representative substrates from each group. Pictures of the rest of substrates are shown in Appendix A. SEM images taken in AEE mode allow to scan graphene with the table current. Thanks to that, places covered and non-covered with graphene are well visible. Characteristic darker lines, well visible for graphene on Si/SiO_2_, kapton, PEN and even on Al_2_O_3_, are the places where the discontinuities of the graphene layer are present. SEM images do not allow to indicate the cause of graphene’s heterogeneity. Graphene can be mechanically damaged during the transfer process. [13]. Defects in graphene may also appear at the stage of its manufacturing. Graphene HSMG is a polycrystalline material and the layer discontinuities may result from the mismatch of individual graphene plates that are lifted on the liquid matrix during the production process. [14]. The SEM image of graphene on quartz glass indicates the existence of areas where multilayers exist.

The Raman spectroscopy is a very useful method to analyze the quality of graphene. The G (~1585 cm^−1^) and 2D (~2700 cm^−1^) peaks are characteristic for graphene Raman spectrum. The G band is the only band coming from a normal first order Raman scattering in graphene [15,16,17]. The 2D peak has symmetrical shape and can be fitted by one Lorenzian. The intensity of 2D peak decreases when defects appear. The full width at half maximum (FWHM) of 2D band for graphene monolayer is about 25 cm^−1^ and widens when the amount of graphene layer increases or when defects occur. The ratio of 2D to G peak intensity (I_2D_/I_G_) amounts to 4 for pristine graphene and decreases when defects occur [18,19]. D (~1350 cm^−1^) and D’ (~1620 cm^−1^) bands are activated by single-phonon intervalley and intravalley scattering process. Presence D and D’ peaks on Raman spectrum confirm the defect occurring [20].

To test the graphene quality, Raman map I_2D_/I_G_ was made (Appendix A). The ratio of intensity of 2D and G peaks demonstrates that the graphene was in homogeneity. The sample consists of regions with different amounts of graphene layer (from one—FWHM = 32 cm^−1^—to three (the fitting of this peak is presented in Appendix A). To check the difference between reaction of mono and multilayer graphene on UV irradiation, the Raman spectra were taken before and after irradiation (Figure 3). Peaks D and D’ prove that the defects are present in tested graphene [20]. Comparing the Raman spectra, it can be noticed that UV irradiation of graphene causes increases of defect. This is indicated by the increase of D and D’ bands’ intensity as well as the decrease of the 2D peak intensity. It can be clearly seen that the monolayer is more sensitive to UV irradiation. Significant increase in the ratio of D to G peak intensity (I_D_/I_G_) (Table 1) can be observed. This is in agreement with the results of G. Imamura et al. [4]. According to the methodology presented by Concado et al. [21] the distance between defects (L_D_) was calculated (Table 1). The calculations indicate that there is significant reduction in the distance between point defects in graphene after UV irradiation. Considering the increasing of the ratio of D to D’ peak intensity (I_D_/I_D’_) after UV irradiation, it can be concluded that UV radiation changes the nature of defects. Referring to the values obtained for the graphene monolayer given by A. Eckmann et al. (where the ratio: I_D_/I_D’_ about 3.5 is characteristic for boundary-like defect, I_D_/I_D’_ about 7 is typical for vacancies, I_D_/I_D’_ about 13 associated with sp^3^ hybridization) it can be concluded that UV radiation can cause the formation of sp^3^ bonds [22]. As it was described earlier by G. Imamura et al., the authors suggest that the reason of defect formation in graphene due to UV irradiation is caused by the formation of sp^3^-like bonds on the bounder of SiO_2_ and graphene [5].

The sensitivity of graphene to UV was measured by the change in resistance value. The results obtained for various types of substrates are shown in Figure 4. The relative change in resistance displayed on the Y axis is given by the formula
∆R/R (%) = (R_time_−R_initial_)/R_initial_ × 100%
where R_initial_ and R_time_ are the sensor resistance change within time in response to UV irradiation.

The tested samples were characterized by different values of initial resistance from approx. 1.6 kΩ to approx. 5.7 kΩ (Table 2). The reason for such a large discrepancy in the resistance value can be both the structural defects of technological graphene (defects of the crystalline lattice, growing next layers), as well as mechanical damages occurring during the transfer. It would be expected that such large differences in initial resistance will affect the electrical response of graphene to UV irradiation. The obtained results do not confirm the existence of such a relation between the initial value of graphene resistance and the electrical response of the tested sensors. The samples with low initial resistance: Si/SiO_2_ and PEN are characterized by a completely different UV response: Si/SiO_2_, 92%; PEN, 23%, while ceramics with high initial resistance exhibit sensitivity at 55%.

On the graph of resistance changes, the relationship between the final substrate and the character of changes in graphene response to UV is clearly visible. Graphene transferred on polymeric substrates is characterized by greater dynamics of reaction and reaches the maximum value of the resistance faster (Table 2). In the progress of resistance changes for these materials, three stages can be observed: (1) dynamic growth, (2) stabilization, (3) slow decrease. The final resistance values obtained for the remaining substrates indicate slow reaching of the stabilization stage by the tested sensor.

The influence of UV irradiation on the carriers’ concentration and their mobility for all substrates was also examined. For each of the substrates, a decrease of carriers’ concentration was observed with simultaneous increase in their mobility due to irradiation (Figure 5). The graphene monolayer, because of contact with both the surrounding atmosphere and the substrate, behaves as a p-type semiconductor. UV irradiation reduces the number of holes and increases the resistance of graphene. Similar behavior of graphene has already been described in the literature, but its cause remains an open question. The reaction of graphene to UV irradiation is explained by: adsorption and desorption of water and oxygen from the surface of graphene [6,22,23], interaction of graphene with the substrate [5] and overlapping the two above-mentioned interactions [1,24].

To check which interactions with graphene are predominant, an additional test was performed. Graphene on substrates that are resistant to high temperatures (Si/SiO_2_, quartz glass and Al_2_O_3_) was annealed at 500 °C under 2∙× 10^−3^ Pa. That allowed to clean the graphene surface and get rid of water trapped between graphene and the substratum. After annealing, the samples were kept under ambient conditions for 24 h. This time was enough to adsorb the water vapor on the graphene. Then the samples were irradiated according to the previously adopted scheme.

For all tested samples, an increase in resistance after annealing was observed: Si/SiO_2_, 2 kΩ; quartz glass, 5.14 kΩ and Al_2_O_3_, 50 kΩ. The reason for the increase in resistance may be graphene damage during heating through the release of water. The release of a large amount of water trapped in the pores of the Al_2_O_3_ was a very dynamic process and led to a significant damage of graphene. The samples reacted less dynamically to UV irradiation after annealing. There was only a few percent increase in UV resistance. The progress of resistance changes for all tested samples had a similar character to the unheated polymer substrates (Figure 6) (response stages: dynamic growth, stabilization, slow decline). The highest electrical response was obtained for the Al_2_O_3_ substrate and was the lowest for Si/SiO_2_.

A decrease of carrier mobility was observed. It may be associated with the increase of amount of defect created in graphene during annealing. Differences were also observed in change of carrier concentration. A slight decrease in concentration after irradiation was observed for Si/SiO_2_. For the remaining substrates a few percent carrier concentration increase was observed (Figure 7).

The results indicate the relationship between the sensitivity of graphene to UV irradiation and the amount of water trapped under graphene during the transfer. Quartz glass and Si/SiO_2_ exhibit hydrophilic properties, while polymeric substrates and ceramics are hydrophobic. Depending on wettability of the surface, the amount of water enclosed under the graphene surface varies. It is greater for hydrophilic materials and smaller for hydrophobic materials, which results in a difference in the nature of changes in the electrical response. An exception is ceramics, which despite the hydrophobic properties have the ability to accumulate more water in the pores. Because of that, their electrical response is similar to hydrophilic materials. Annealing of the samples allows to get rid of water trapped under graphene, making graphene much less sensitive to UV radiation.

The reason for the highest electrical response obtained for Al_2_O_3_ is water adsorbed in the pores. The water is not completely removed during the heating. The lowest sensitivity of graphene transferred on Si/SiO_2_ indicates the almost complete removal of water below the graphene. Trying to explain the nature of phenomena affecting the electrical response of the sensor under the UV irradiation, it can be concluded that the dynamic increase in resistance in the initial phase of irradiation is related to the purification of graphene from water. The drop of resistance observed in the last phase is caused by the interaction of graphene with the substrate. The stabilization stage is a temporary balance between the two above-mentioned processes.

## 4. Conclusions

The subject of the research was the influence of the final substrate on sensitivity of graphene to UV irradiation. To determine this correlation, the electrical response of graphene was measured. The most important feature that has an influence on sensitivity of graphene to UV is the ability of the substrate to accumulate the water on the surface. The high amount of water adsorbed on the final substrate surface affects both the dynamics of the electrical response of the sensor and the progress of the character of resistance changes. Getting rid of water trapped on the graphene-substrate boundary results in a decrease in the sensitivity of graphene to UV irradiation. Summarizing, the final substrate used in UV sensors should be characterized by hydrophilic properties to provide the appropriate amount of water on the boundary.

## Figures and Tables

**Figure 1 materials-12-03949-f001:**
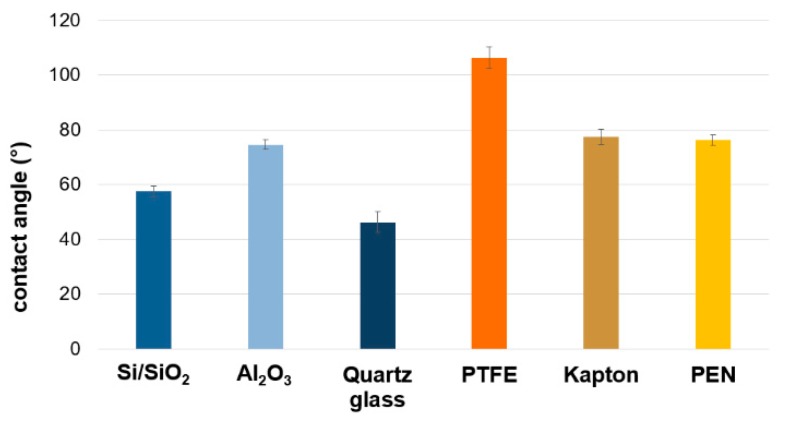
Contact angle of used substratum.

**Figure 2 materials-12-03949-f002:**
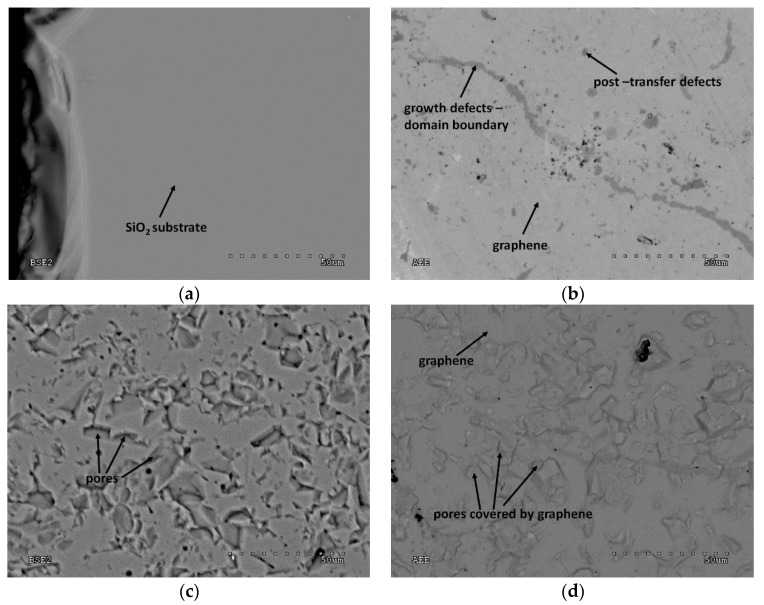
State of the substrates’ surface before transfer and the quality of graphene on final substratum. (**a**) Si/SiO_2_, (**b**) graphene on Si/SiO_2_, (**c**) Al_2_O_3_, (**d**) graphene on Al_2_O_3_.

**Figure 3 materials-12-03949-f003:**
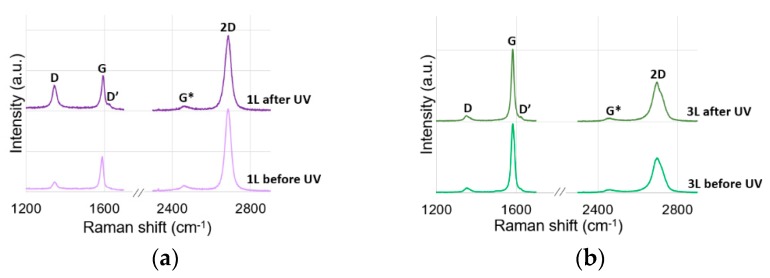
Raman spectra before and after exposure to UV irradiation for (**a**) monolayer (1 L) and (**b**) trilayer (3 L) region of graphene.

**Figure 4 materials-12-03949-f004:**
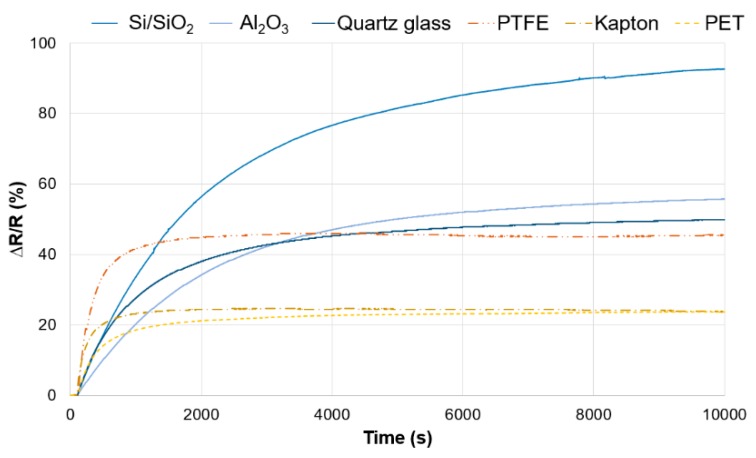
Relative change in resistance under UV irradiation for graphene deposited on various substrates.

**Figure 5 materials-12-03949-f005:**
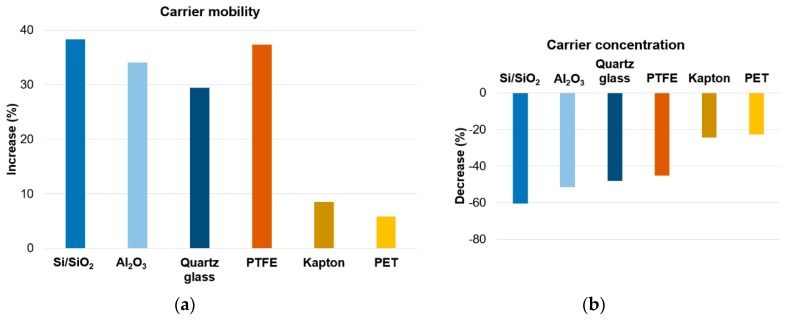
Change of mobility (**a**) and carrier concentration (**b**) under UV irradiation.

**Figure 6 materials-12-03949-f006:**
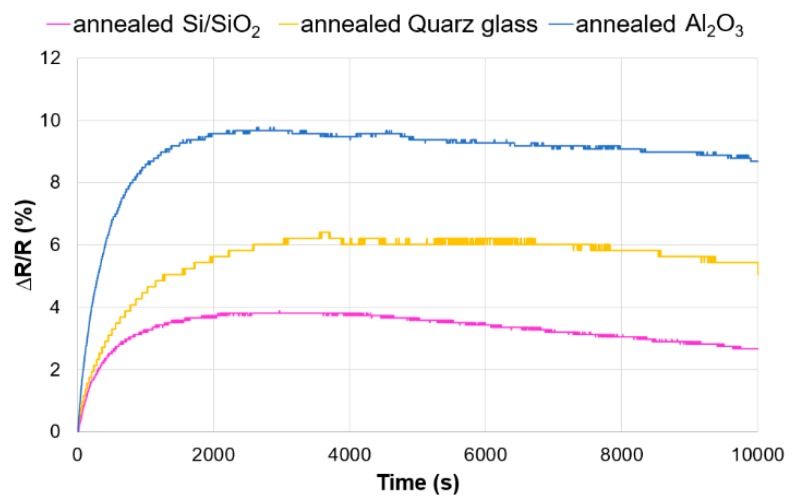
Relative change in resistance under UV irradiation for annealed samples.

**Figure 7 materials-12-03949-f007:**
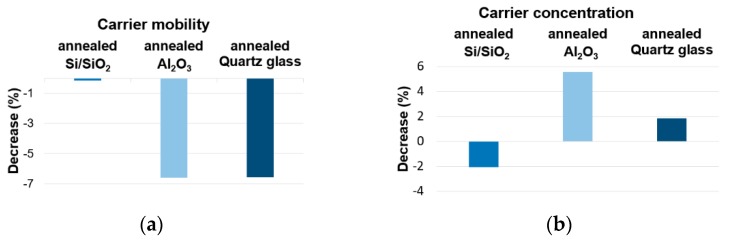
Change of motility (**a**) and carrier concentration (**b**) under UV irradiation for annealed samples.

**Table 1 materials-12-03949-t001:** Ratios of typical peak calculated on the basis of Raman spectra deconvolution and the distance between point defects in graphene before and after irradiation.

Graphene Type	-	I_2D_/I_G_	I_D_/I_G_	L_D_ (nm)	I_D_/I_D’_
monolayer	before UV	2.56	0.21	29	4.52
	after UV	2.32	0.69	16	9.32
trilayer	before UV	-	0.06	55	1.37
	after UV	-	0.07	52	2.49

**Table 2 materials-12-03949-t002:** The values of resistance for graphene transferred on final substrate during UV irradiation test.

Resistance	Substratum					
	Si/SiO_2_	Al_2_O_3_	Quartz Glass	PTFE	Kapton	PEN
**R_0_ (Ohm)**	1687	5770	3990	3337	3850	2300
**R_F_ (Ohm)**	3250	9370	5980	4860	4760	2846
**R_max_**	3250	9370	5980	4870	4800	2847
**R_max_ Time (s)**	9873	9950	9501	3158	2457	2843

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
