# Peer review of "Influence of the Interactions at the Graphene–Substrate Boundary on Graphene Sensitivity to UV Irradiation"

_materials, 2019, doi:10.3390/ma12233949_

Round 1

Reviewer 1 Report

There is certainly merit in this study but the manuscript needs improvement prior to publication. There are many typographical errors and a lack of acronym definitions. There is insufficient detail in the experimental sections (for example what was the pressure in the vacuum chamber during the anneal). Insufficient attention is given to the discussion over the surface and sample preparation which leads to lack of credibility regarding the conclusions for example even in line 45 desorption is discussed but not what is desorbed.   

Author Response

Dear Reviewer:
Please find answers for your review in attachemen.

Reviewer 2 Report

This manuscript cannot accept in current form. Author needs to revise this manuscript carefully.

Raman peaks are not clear. Author need to show D, G 2D band. Follow this below manuscript.

     “Raman spectrum of graphene with its versatile future perspectives”

AFM images  require for both monolayer and trilayer graphene.

Figures are not clear. Author should provide very clear image. Images are very poor, it is difficult to go through it.

Author Response

(The authors gave the same response as above.)

Round 2

Reviewer 1 Report

The paper on the influence of the interactions at the graphene - substrate boundary on graphene sensitivity to UV irradiation, is much clearer and easy to read with the English also being improved. I would therefore recommend publishing this interesting and useful research.

Reviewer 2 Report

Your manuscript is accepted.